# Social learning strategies modify the effect of network structure on group performance

Daniel Barkoczi[1] & Mirta Galesic[1,2]

The structure of communication networks is an important determinant of the capacity of teams, organizations and societies to solve policy, business and science problems. Yet, previous studies reached contradictory results about the relationship between network structure and performance, finding support for the superiority of both well-connected efficient and poorly connected inefficient network structures. Here we argue that under-standing how communication networks affect group performance requires taking into consideration the social learning strategies of individual team members. We show that efficient networks outperform inefficient networks when individuals rely on conformity by copying the most frequent solution among their contacts. However, inefficient networks are superior when individuals follow the best member by copying the group member with the highest payoff. In addition, groups relying on conformity based on a small sample of others excel at complex tasks, while groups following the best member achieve greatest performance for simple tasks. Our findings reconcile contradictory results in the literature and have broad implications for the study of social learning across disciplines.

[1] Center for Adaptive Behavior and Cognition, Max Planck Institute for Human Development, Lentzeallee 94, Berlin 14195, Germany. [2] Santa Fe Institute, 1399 Hyde Park Road, Santa Fe, New Mexico 87501, USA. Correspondence and requests for materials should be addressed to D.B. (email: barkoczi@mpib-berlin.mpg.de).

The trade-off between exploration and exploitation lies at the heart of many problems faced by individuals, groups and organizations, who often need to decide whether to search for new, potentially better solutions (for example, a technology, social institution or a business strategy) or keep using an existing solution that works well[1–6]. The right balance between exploration (searching for superior novel solutions) and exploitation (reaping benefits of existing solutions) is thought to be essential for adaptive behaviour in humans and other animals[4,5,7]. When individuals interact through social learning to solve problems collectively, this trade-off is manifested in the balance between innovation through individual learning from the environment and the imitation of existing solutions in the population[8–12]. Innovation (exploration) is essential both for tracking changes in the environment and for introducing novelty in the population, while imitation (exploitation) serves the purpose of diffusing good solutions to increase group-level performance[13].

How do different social learning strategies affect the balance of exploration and exploitation, and the resulting performance? And how do these strategies interact with the social or organizational network in which learning takes place? We address these two questions by modelling different social learning strategies as algorithms composed of three cognitively plausible building blocks, previously studied in the literature on individual decision-making in non-social contexts: rules that guide information search, stopping search and making a decision[14]. We study how groups of individuals using different strategies perform in task environments characterized by different levels of complexity, while embedded in social networks varying in structural properties that have been shown to affect the ease of information flow in communities[8,15–20].

We make two contributions. First, we demonstrate how the building blocks of social learning strategies can lead to strikingly different exploration–exploitation patterns and, as a result, to different levels of performance in simple and complex task environments. Second, we clarify and reconcile seemingly contradictory results in the literature by showing how social learning strategies and network structure interact to affect group performance. Specifically, a number of studies have found that network structures that promote slower information diffusion (are less efficient) enhance group performance because they lead to higher levels of exploration and increase the chance of finding better solutions in the population[8,15,18,21]. In contrast, a recent study, focusing on the same question, came to the opposite conclusion, finding that networks promoting faster information flow (those that are more efficient) lead to better performance[16]. Here we show that one can obtain both results for the same type of problem-solving task. We argue that answering the question of how network structure affects performance requires studying how it interacts with the social learning strategies used by individuals.

We develop a model of problem solving, where a group of individuals are repeatedly searching for solutions that improve group-level performance. We follow several authors in modelling this problem as search on rugged landscapes[15,16,18,22,23]. The main difficulty encountered by problem solvers searching such environments is the presence of several local optima (peaks) from which it is difficult to find better solutions. As a result, groups face the challenge of finding good solutions without getting stuck in local optima. We focus on two different environments, a simple one with a single optimum and a complex one with several locally optimal solutions (see Methods for further details).

In separate simulations, we vary the structural properties of the communication networks in which agents are embedded. At the two extremes, we consider a 'fully connected network', where each individual is connected to everyone else in the population,

and a 'locally connected lattice', where individuals are connected to their $d$ immediate neighbours. In addition, we consider eight network structures that were proposed in a recent study focusing on the relationship between network structure and group performance[16]. Taken together, these networks cover a broad spectrum of possible structures and include all of the networks that were studied in Lazer and Friedman[15], Mason and Watts[16], and Derex and Boyd[18], the recent studies that reached incompatible results about the role of network structure on group performance.

For each network structure and task environment, we assume a group of 100 agents exploring the problem space through social and/or individual learning. We formalize the social learning strategies as algorithms composed of three basic building blocks: rules that guide information search, stopping search and making a decision[14]. Social learning strategies differed in the number of individuals looked up before stopping search ($s = 3$ or $s = 9$) and the decision rule (best member, conformity). See Methods for further details.

For each social learning strategy, we assume that individuals observe whether the socially acquired solution produces a higher payoff than their current solution. If yes, they switch to the socially acquired solution; otherwise they engage in individual learning. Therefore, these strategies can be seen as a form of 'critical' social learning[24].

Following several authors[15,25,26], we model individual learning as a hill-climbing strategy, which explores the landscape by modifying a single digit in the current solution and observing whether it produces a higher payoff. If yes, individuals switch to the alternative solution; otherwise they keep their current solution. As baseline strategies, we consider a group of pure individual learners, who engage only in exploration (individual learning) and a group of pure social learners, who engage only in exploitation through copying a single individual at random and adopting the individual's choice if it has a higher payoff (random copying).

On each time step, individuals first engage in social learning (as described above) and, conditionally, switch to individual learning. We iterate the procedure for $t = 200$ time steps, and record the average payoff in the population on each time step separately for each combination of strategy, network structure and task environment. Results reported are averaged across 1,000 repetitions.

As we show next, inefficient network structures outperform efficient ones when individuals rely on the best member strategy, while efficient networks outperform inefficient ones when individuals rely on the conformity strategy. In addition, groups relying on conformity based on a small sample of other individuals excel on complex tasks, while groups following the best member achieve greatest performance for simple tasks.

## Results

**Performance of different social learning strategies**. Figure 1 shows the average payoff achieved by each strategy over time for two different environments: a simple one with a single optimum ($N = 15$, $K = 0$, Fig. 1a) and a complex one with several local optima and a global optimum ($N = 15$, $K = 7$, Fig. 1b) (see Methods for more details). Results are qualitatively the same for all other values of $K > 0$ and $s$ (see Supplementary Table 1 for results for other values of $K$; Supplementary Fig. 1 for a plot of performance variability across repetitions; Supplementary Table 2 and Supplementary Note 1 for results for other sample sizes). Here we focus on the performance of different social learning strategies in a fully connected network, and discuss different network configurations in the next section.

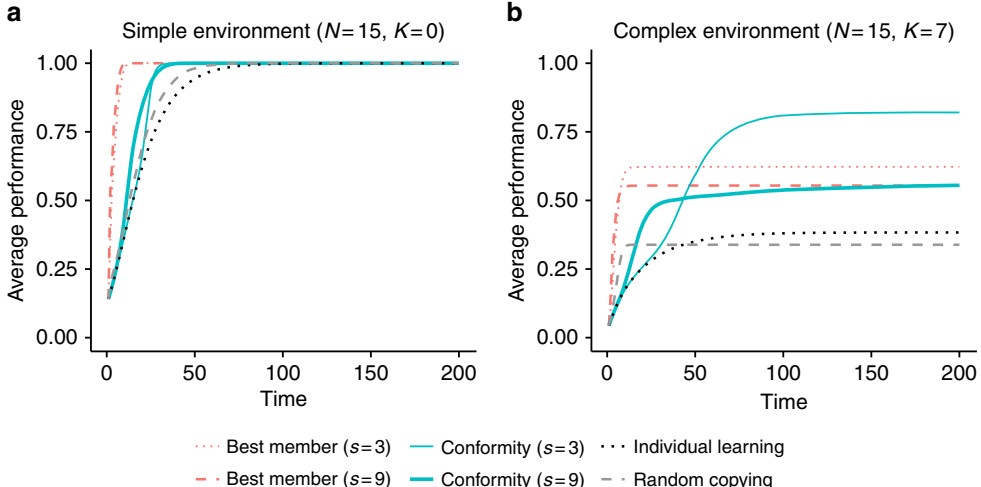

**Figure 1 | Performance over time for different strategies in a fully connected network.** *N* denotes the number of components of the system and *K* represents the number of interdependent components (see Methods); *s* stands for sample size. (**a**) Simple environment with a global optimum and (**b**) complex environment with multiple local optima. Red dotted lines: best member (*s* = 3); red dashed lines: best member (*s* = 9); turquoise thin lines: conformity (*s* = 3); turquoise thick lines: conformity (*s* = 9); black dotted lines: individual learning; grey dashed lines: random copying. Based on *n* = 100 agents and 1,000 repetitions.

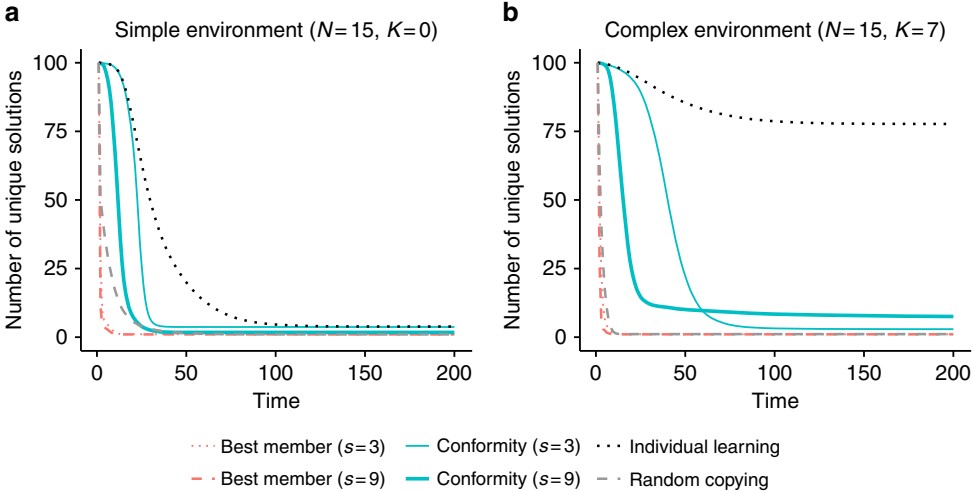

**Figure 2 | Number of unique solutions in the population over time.** *N* denotes the number of components of the system and *K* represents the number of interdependent components; *s* stands for sample size. (**a**) Simple environment with a single optimum and (**b**) complex environment with multiple local optima. Red dotted lines: best member (*s* = 3); red dashed lines: best member (*s* = 9); turquoise thin lines: conformity (*s* = 3); turquoise thick lines: conformity (*s* = 9); black dotted lines: individual learning; grey dashed lines: random copying. Based on *n* = 100 agents and 1,000 repetitions.

**Performance in simple environments**. In the simple environments (where *K* = 0), all strategies eventually find the global optimum; however, strategies differ in the time they need to converge to this optimum (Fig. 1a). Strategies relying on the best member lead to the fastest convergence, followed by the conformity strategies and the pure individual and social learning strategies. Because simple environments are dominated by only one optimum, the tension between exploration and exploitation is not very pronounced, since eventually every individual will end up finding the best solution. The fact that all individuals converge on the same solution can also been seen from Fig. 2a, which shows the number of unique solutions in a population over time. The number of unique solutions converges to 1 for all strategies, which in the case of simple environments happens to be the global optimum.

**Performance in complex environments**. Figure 1b shows two striking results. First, the small-sample versions of both strategies outperform their large-sample versions. This occurs because small samples provide noisy information about the frequency of different solutions in the population. This noisy information reduces the chance that individuals can find good solutions early on and, as a result, makes individuals engage in higher levels of exploration. This in turn increases the chance that over time they will find better solutions.

Second, the conformity strategy relying on small samples (*s* = 3) converges to the highest long-run outcomes, outperforming the best member strategies by a large margin. Best member strategies reach the highest short-run outcomes, but they quickly drive the whole population towards locally optimal solutions, from which individual exploration is no longer able to find better

 

solutions (see also refs 7,15,25). As a result, the whole population gets stuck in an inferior state. This can also be seen from Fig. 2b, which shows that the best member strategies converge to a single unique solution. In contrast, the conformity strategies converge more slowly (and to multiple solutions), leading to higher level of exploration while still allowing the infrequent but superior option to diffuse through the population.

The superiority of the small-sample conformity ($s = 3$) strategy stems both from the heightened levels of exploration and from its capacity for diffusing rare but superior solutions through the population (see also ref. 27). That heightened exploration alone is not enough can be seen from the fact that small-sample conformity outperforms pure individual learning, which relies exclusively on exploration. This pattern of results was replicated in all complex landscapes ($N > K > 0$, see Supplementary Table 1).

With regards to the pure learning strategies, random copying engages in high levels of exploitation in the beginning and drives individuals to a single, locally optimal solution (as can be seen from Fig. 2). However, since it is a pure social learning strategy and engages only in exploitation, it can only disseminate information that is already present in the population. It outperforms individual learning initially, but reaches worse performance in the long run. On the flip side, pure individual learning engages only in exploration, but cannot spread the good solutions through the population. The low performance of the two pure learning strategies demonstrate the need to balance exploration and exploitation[4].

Taken together, these results indicate that different social learning strategies lead to different patterns of explorative and exploitative behaviour over time. The extent to which different strategies prove useful depend crucially on their building blocks (search, stopping and decision rules). Strategies using the best member decision rule lead to high levels of exploitation and drive the population towards local optima. Strategies using the conformity decision rule promote higher levels of exploration and can enable the population to find higher-payoff solutions when relying on small samples. We replicate the same finding in changing environments (Supplementary Fig. 2; Supplementary Note 2).

**Network structure and social learning strategy interact**. We have seen that different strategies can achieve remarkably different performance within the same network structure. At the same time, different network structures are also known to affect performance by changing the relative use of exploration and exploitation in the group[8,15,16,18,21]. How does network structure interact with the social learning strategies?

Previous studies reached contradictory results. For example, Lazer and Friedman[15] used an agent-based simulation to compare a 'fully connected network' with a 'locally connected lattice' and found that the 'locally connected lattice' outperformed the 'fully connected network' in the long run (see also refs 8,18). This result implies that inefficient networks (that lead to slower information spread) achieve better outcomes. Mason and Watts[16] report a behavioural experiment with eight different networks (Supplementary Table 3; Supplementary Fig. 3; Supplementary Note 3). In contrast to Lazer and Friedman[15], they find that efficient networks (that are faster at spreading information in the population) outperform inefficient networks.

What is driving this difference in results? Here we show that both results can be obtained depending on the social learning strategies that individuals use in a given network. We study 10 different networks (Supplementary Table 3; Supplementary Fig. 3). We focus on the two best performing social learning strategies in complex task environments, the

best member and conformity strategies with small samples ($s = 3$).

Figure 3a shows the average payoff achieved by groups in different networks. The left panel shows the average performance of the efficient and inefficient networks when individuals use the best member strategy, while the right panel shows the same performance when individuals rely on the conformity strategy. The shaded regions show the area between the best and worst performing networks in each category. Results in the left panel replicate the findings of Lazer and Friedman[15], who find that inefficient networks outperform efficient networks. This is expected given that their simulated agents relied on the best member strategy. Results in the right panel show the opposite result, with efficient networks outperforming inefficient networks. This is in line with the results of Mason and Watts[16], who found the same result, albeit in a different type of landscape (we replicate these same findings using the landscape of Mason and Watts[16], and report the results in the Supplementary Information (Supplementary Fig. 4; Supplementary Note 4)). Our results indicate that this finding would be expected if participants used a strategy similar to conformity. By comparing the two panels, we can also see that the conformity strategy outperforms the best member strategy in each network. The same conclusions can be drawn from the bottom row of Fig. 3, which shows average performance at the final time step ($t = 200$) for each network.

We also examine the relationship between the diameter of a network and the average payoff achieved by individuals. Higher diameters indicate less efficiency at spreading information. Figure 4 shows that the relationship between diameter and average performance is positive for the case when individuals rely on the best member ($s = 3$) strategy and negative when individuals rely on the conformity ($s = 3$) strategy, confirming that network efficiency has opposite effects depending on the strategy being used (the relationship between clustering coefficient and average performance shows the same pattern of results). Note, however, that our analyses are based on an uneven number of diameter values, since some of the networks have the same diameter.

Our findings demonstrate that both efficient and inefficient networks can lead to superior performance, depending on the social learning strategies used by individuals in the group. They suggest that network structure and social learning strategies jointly affect the levels of exploration and exploitation in the population. If both strategy and network promote high levels of the same activity (either exploration or exploitation), performance is likely to drop; however, if network and strategy promote opposite behaviours, performance is likely to rise.

## Discussion

We asked two questions. First, how do different social learning strategies affect behaviour and performance in simple and complex task environments? We found that the best member strategies reach the highest performance in simple task environments, but weak conformity, achieved by relying on small samples, ensures the highest long-run outcomes when task environments are complex. The intuition underlying these results is the following. The best member strategies are fast at diffusing useful information and, therefore, quickly drive the population towards locally optimal solutions. The conformity strategy leads to slower convergence and thereby allows the population to explore more and eventually find solutions that have higher payoffs. Small samples have a similar effect and help both strategies in complex environments.

Second, how do these strategies interact with network structure? Our results indicate that efficient networks promoting

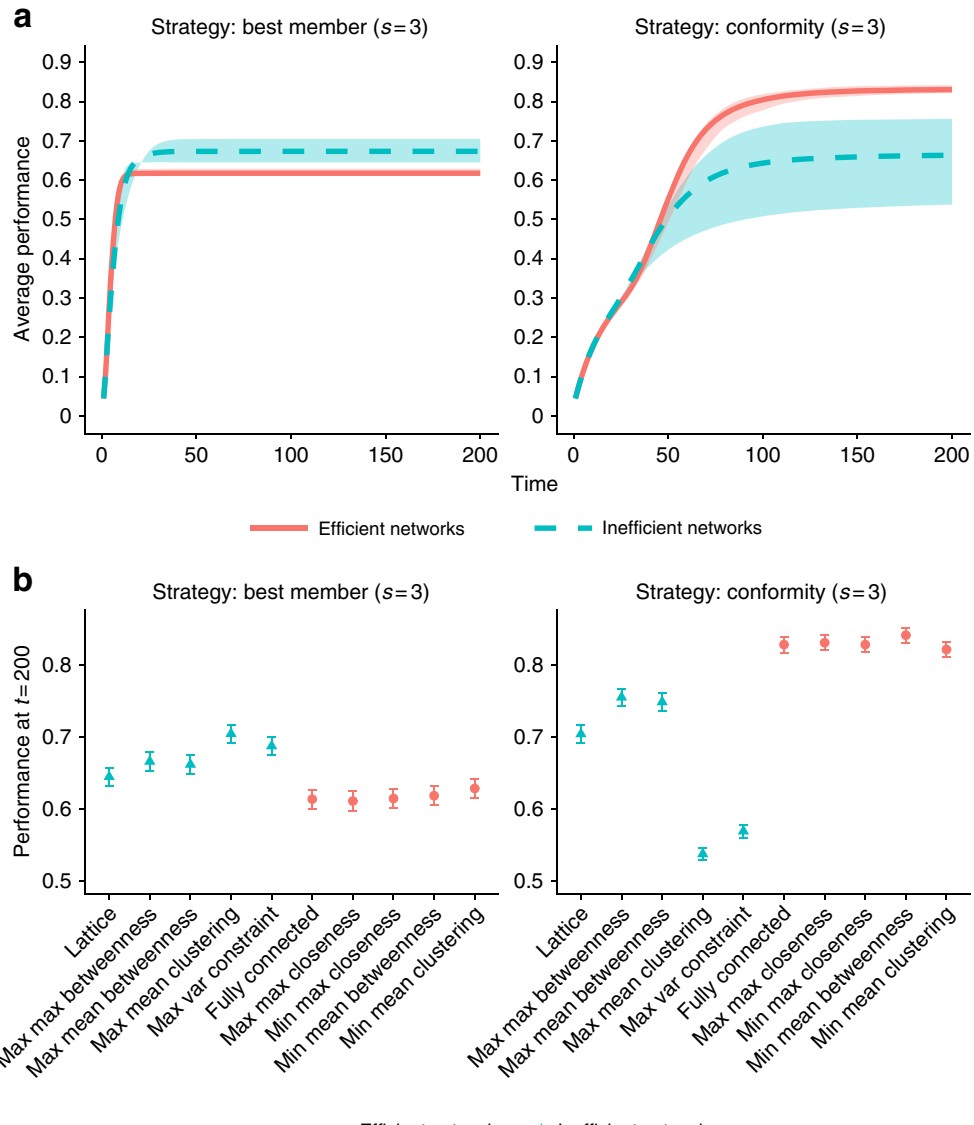

**Figure 3 | Performance of different networks as a function of social learning strategy. (a)** Group performance averaged across efficient (red solid lines) and inefficient (turquoise dashed lines) networks. Shadings around the lines show the region between the best and worst performing network in each category. **(b)** Average performance at the final time step ($t = 200$) for each network. Error bars show $+ - 2$ s.e. of the mean. Left panels: individuals rely on the best member ($s = 3$) strategy; right panels: individuals rely on the conformity ($s = 3$) strategy. Inefficient networks outperform efficient networks when individuals rely on the best member strategy. Efficient networks outperform inefficient networks when individuals rely on the conformity strategy. Results in the left panel replicate the findings of Lazer and Friedman[15] while results in the right panel are in line with Mason and Watts[16]. Based on $n = 100$ agents and 1,000 repetitions.

faster information diffusion outperform the inefficient networks when individuals use the conformity strategy. However, the opposite is the case when individuals use the best member strategy: here inefficient networks outperform efficient ones. This shows that group performance depends both on the network structure individuals are embedded in as well as the social learning strategies they use. We used this insight to clarify and reconcile seemingly contradictory findings from the literature, by showing that both well-connected and less well-connected networks can be beneficial for the same task, depending on the social learning strategies used by individuals[8,15,16,18,21]. Our results are in line with recent analyses showing that, for complex tasks, there is a trade-off between the probability of adopting others' solutions and network connectivity[28,29]. We show that such trade-offs can be achieved by cognitively plausible social

learning strategies that are widely observed among human and other animals[30]. Recently, Shore *et al.*[20] showed that different network structures can be better for different types of tasks involved in problem solving. Here we show that social learning strategies can change the performance of different networks for the exact same task.

Our results replicate the findings of both Lazer and Friedman[15], who studied the best member strategy in agent-based simulations and found superiority of inefficient networks, as well as the findings of Mason and Watts[16], who conducted behavioural experiments and found that more efficient networks are superior. While we do not know the exact strategies that participants employed in the latter experiments, Fig. 4 in Mason and Watts[16] and the surrounding discussion suggests that their participants where disproportionately more

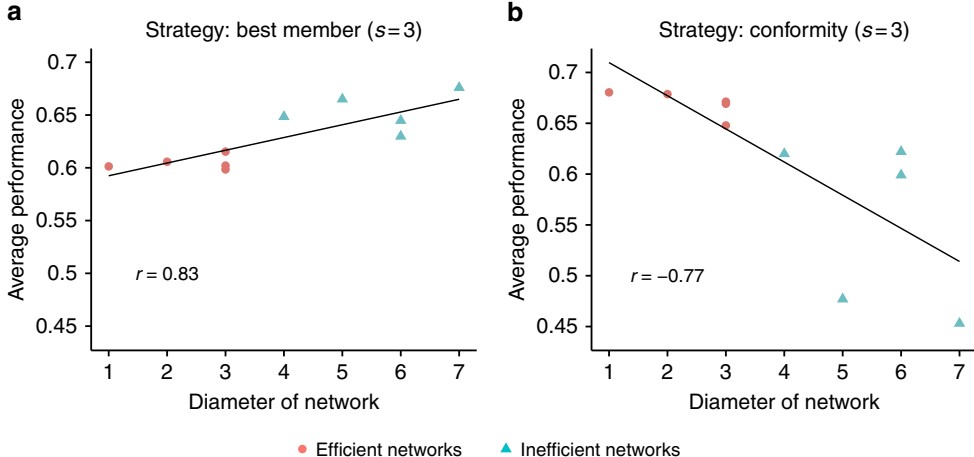

**Figure 4 | Relationship between network diameter and average performance.** (**a**) Individuals rely on the best member ($s=3$) strategy; (**b**) individuals rely on the conformity ($s=3$) strategy. Red circles indicate efficient networks and turquoise triangles indicate inefficient networks. Straight lines show best linear fits. The relationship between diameter and performance is positive when individuals rely on the best member strategy (Pearson correlation coefficient $r=0.83$) and is negative when they rely on the conformity strategy ($r=-0.77$).

likely to copy a solution if two or all three group members exhibited it, compared with a solution found by only one of the three other members. This finding indicates that they relied on some form of frequency-dependent social learning, similar to our conformity strategy[31]. Mason and Watts[16] also found that individuals in efficient networks explored more than those in inefficient networks, suggesting that network efficiency on its own should not lead to premature convergence on local optima. Our results agree with this finding and suggest that the social learning strategy used by individuals will also influence the speed of convergence to different optima.

Overall, our findings provide a novel perspective on the relationship between network structure and group performance, and raise a number of issues that could be tested in future empirical studies. While previous experiments have focused on fixed groups where individuals have access to the choices of all of their connections, the question of how individuals decide whom to copy and how many individuals they sample remains unaddressed. Studies of this question could also provide insights into how people decide to form network connections and how network structure evolves during the learning process. For the sake of clarity and our goal of reconciling existing results in the literature, we focused on groups that rely on a single form of social learning in fixed network structures. Future work should investigate how different mixtures of social learning strategies both within and across individuals affect performance in networks[32,33], as well as how network structure affects the selection of different social learning strategies.

It is well known in the evolution of social learning literature that the performance of a strategy depends crucially on the types and frequencies of other strategies in the population[11,12]. In this study we assume that groups are using a single strategy. While this is clearly a limitation of the present study, we believe it is justifiable for two reasons. First, studying the dynamics of strategies in isolation is a necessary first step towards understanding how they might affect other properties of the system (here, network structure). Second, the results gained from studying populations with mixed strategies can always be modified by the inclusion or removal of some strategies from the mixture. A key challenge for future research will be to empirically estimate the frequencies and forms of different social learning strategies to inform theoretical models about the variety and frequency of social learning strategies in real-world

groups[33,34]. Finally, future studies could employ real-world networks from large-scale organizations or online platforms.

Our study has broad implications for organizational learning, technological innovation, cultural evolution and the diffusion of innovations. Research on technological innovation has highlighted the combinatorial nature of innovation with most new inventions being re-combinations of existing technologies[35,36]. Much of this research has focused on how innovation occurs, whereas there has been very little attention devoted to the co-evolution of innovation and the simultaneous diffusion of these innovations. We identify situations where imitation can both help and hinder the development of technological innovation. In addition, most studies of exploration and exploitation in organizations focus on how to design the external environment to make groups more adaptive[8,15,25,37]. Our results highlight that it is also important to consider the social learning strategies used by individuals and organizations, and show that interventions aimed at changing the social environment while disregarding these strategies might not produce the desired effects.

## Methods
**Generating the task environment.** To create a multi-peaked environment, we use the NK model[26], which is a 'tunably rugged' landscape determined by $N$, the number of components that make up each solution, and $K$, the number of interdependencies between these components. $N$ and $K$ together determine the structure of the task environment where different solutions in the space have different payoffs (see Supplementary Fig. 5 for an illustration of the landscapes). To construct the task environment, we represent each solution in the environment by an $N$-length vector composed of binary digits, leading to a total of $2^N$ possible solutions in the task environment. The payoff of each solution is calculated as the average of the payoff contributions of each element. The payoff contribution of each element is a random number drawn from a uniform distribution between 0 and 1. In the case of $K=0$, a simple average of the $N$ elements is taken: $\left(\frac{1}{N}\right)\sum_{i=1}^{N} N_i$, whereas with $K>1$, individual payoff contributions are determined by values of the $K-1$ other, interdependent elements, that is, $f(N_i|N_iN_{i+1},...,N_k)$, where $f()$ is the payoff function and the total payoff is $\frac{1}{N}\sum_{i=1}^{N}f(N_i|N_i,N_{i+1},...,N_k)$. Which of the $K-1$ other components a given element is interdependent with is determined randomly. In other words, when $K=0$, changing any single element of the solution will affect only the contribution of that element, whereas when $K>0$, changing a single element will change the payoff contribution of the $K-1$ other elements. When $K=0$, exploration of solutions through the modification of single components can prove effective, but as $K$ increases, local exploration becomes less and less effective[7].

Following several authors[15,37], we normalize the payoffs of different solutions by dividing them by the maximum obtainable payoff on a landscape $P_{\mathrm{Norm}}=P_i/\max(P)$. The distribution of normalized payoffs tends to follow a normal distribution with decreasing variance as $K$ increases. This implies that most

solutions tend to cluster around very similar payoff values. Following Lazer and Friedman[15], we use a monotonic transformation, raising $P_{Norm}$ to the power of 8 $((P_{Norm})^8)$ to widen the distribution, making most solutions 'mediocre' and only a few solutions 'very good'.

**Generating the networks.** All networks were assumed to have $n = 100$ nodes and a fixed degree of $d = 19$, except for the 'fully connected' network where $d = n - 1$. This produces the same degree-to-node ratio as in the study of Mason and Watts[16] (more details about all networks are provided in the Supplementary Information). For the 'fully connected network', we assign each node all other nodes as neighbours, leading to $n - 1$ neighbours for each node. For the 'locally connected lattice', we assign each node the $d = 19$ closest consecutive nodes as neighbours, keeping the degree fixed across nodes. To generate the eight additional networks, we follow the procedure described in Mason & Watts[16]. Starting from a random graph with $n = 100$ nodes and a fixed degree of $d = 19$, we perform degree-preserving rewirings. After each rewiring, we record the network measure of interest ((a) closeness centrality, (b) betweenness centrality, (c) clustering coefficient and (d) network constraint) and accept the rewiring only if it maximizes or minimizes the network measure of interest. We continue the rewiring process for 1,000,000 iterations or until we are no longer able to obtain a network with a lower (higher) network measure. We repeat this process 1,000 times to avoid getting stuck in local maxima or minima. The topology of the network in the Supplementary Table 3 indicates the network measure of interest and whether it was maximized or minimized. For example, for the network 'Min mean clustering', we looked at the average clustering in the network, and kept rewiring it until we could no longer find networks with a lower (minimum) average clustering score.

**Simulation procedure.** We assigned random starting solutions to a group of $n = 100$ individuals. This ensured a high level of initial diversity in solutions. On each time step, individuals went through the following steps. First, they applied their social learning strategies consisting of search, stopping and decision rules. Individuals searched randomly among their contacts in the network; stopped searching after looking up the solutions of either a relatively small ($s = 3$) or a relatively large ($s = 9$) sample of other individuals with whom they were connected; made a decision by either selecting the solution of the best performing agent (best member) or the most frequent solution in the sample (conformity). In case each solution was equally frequent, they relied on individual learning (as described below). After applying their social learning strategies, agent compared the socially identified option with their current option and decided to switch if it had a higher payoff. If the socially acquired solution did not produce a higher payoff, they switched to individual learning, which consisted of examining the resulting payoff of modifying a single digit of the current solution and adopting it if it had a higher payoff. These steps were repeated for $t = 200$ time steps for each combination of strategy, task environment and network structure. Results reported were averaged across 1,000 repetitions.

**Data availability.** Simulation code and data files containing simulation output are available at https://github.com/dnlbrkc/social_learning

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

## Acknowledgements

We thank Pantelis Analytis, Robert Boyd, Stojan Davidovic, Rob Goldstone, David Lazer, Winter Mason, participants of the 21st Graduate Workshop in Computational Social Science Modeling and Complexity Workshop held at the Santa Fe Institute and members of the ABC Research group for their comments on earlier versions of the manuscript.

## Author contributions

D.B. designed the research, D.B. performed the research, D.B. and M.G. analysed the data, and D.B. and M.G. wrote the paper.

## Additional information

**Competing financial interests:** The authors declare no competing financial interests.

**How to cite this article**: Barkoczi, D. and Galesic, M. Social learning strategies modify the effect of network structure on group performance. *Nat. Commun.* **7,** 13109 doi: 10.1038/ncomms13109 (2016).

