## [Peer Review File · Nature Communications]

Reviewer #1 (Remarks to the Author)

This is an excellent paper that addresses the utility of different social learning rules. It is very clearly written, and the results are convincing. I recommend that the paper be accepted as is. I have one suggestion that the authors may want to consider. On page 6 the authors describe the random social learning rule as not being biased toward success. Readers who come from the evolution of social learning literature will find this confusing. Random social learners choose a person to learn from (a "model") at random, and then if the behavior that they learn from that person results in a higher payoff, they adopt the socially learned behavior. This is what social learning theorists like Joe Henrich and Richard McElreath call "payoff biased" learning, and it leads to replicator dynamic spread of the higher payoff behavior. It is also the model used by economists interested in social learning (e.g. Karl Schlag) and economists and sociologists who study the diffusion of innovations. The search process is not success biased, but taken over the whole cycle (choose model, learn behavior, decide whether to adopt) the rule is definitely biased toward success because only superior behaviors are adopted. It is very interesting that it fares so poorly compared to a conformist search rule, but the authors might consider adding a few sentences to clarify those who come to this paper with a different conceptual structure.

Reviewer #2 (Remarks to the Author)

This paper presents a simulation of different social learning strategies (e.g. copy the best other agent, or conform to the most common solution amongst n agents) in different fitness landscapes (a simple landscape with a single optimum, or a complex NK multi-peaked landscape) and within different social networks (highly connected vs sparsely connected). It is found that these 3 features interact. In simple landscapes all strategies perform equally well, but in complex landscapes conformity does better because it prevents early convergence on sub-optimal peaks (i.e. increases exploration). Also, conformity does better in efficient networks, but copy-best-agent does better in inefficient networks. The latter effect occurs for the same reason: copy-best's efficiency is reduced by the inefficient network, thus preventing early convergence on sub-optimal peaks. It is claimed that these interactions explain previous apparently-contradictory experimental findings.

Overall I like the study, particularly its claim to resolve conflicting experimental findings. This would be a valuable contribution to the literature. I think it does this only partially however. I have the following concerns/recommendations:

1. It is a problem for me that only in the SI is it mentioned that Mason & Watts actually used a different fitness landscape to the one analysed in the main text. This is a problem because the central claim of the paper is to resolve specifically Mason & Watts' findings with Lazer & Friedman's opposite findings (e.g. from p.9, figure caption: "The left panel replicates the findings of Lazer & Friedman (2007) while the right panel replicates the findings of Mason & Watts (2012)"). Figures S1-S4 show that when Mason & Watts' landscape is used, the effect is much less striking: conformity is still better in efficient networks, but copy-best shows little difference between efficient and inefficient. I would encourage the authors to be explicit about this in the main text, and ideally present these more representative results there rather than in the SI, if the authors wish to claim a link to those earlier studies.
2. Also relating to this comparison, the best-member data presented in Figures 3 and 4 seems to be highly skewed by the lattice outlier. Why is this, and can the authors assure the reader that the effect in the top-left of figure 3 is not simply driven by this outlier? Also, the regression lines in Figure 4 would be far more convincing if there were an equal number of data points for each diameter. Currently there are, for example, three data points for diameter=3, and none for diameter=7 or 8. Please consider increasing the number of diameter values.
3. I would like to see more connection to the evolution of social learning literature, which is highly

relevant here. For example, citation and discussion of models of copy-best and conformity (see Laland, 2004, *Learning & Behaviour*). This has important implications for many of the claims. For example, here the authors are simulating populations in which every member has the same strategy (e.g. all use conformity), measuring performance, then comparing that performance across populations of pure strategies. This is a purely group-level comparison. It assumes that each strategy's performance is independent of the presence of other strategies. The authors cite Rogers (1988) but do not seem to appreciate his result, which is that strategy performance is often not independent of others; they are often frequency dependent. For example, social learners do well amongst large numbers of individual learners because they can scrounge their information at low cost, while social learners do poorly when social learners are common because they cannot track environmental change. The present study cannot make such comparisons because at no point were strategies interacting within the same population. Environmental change is considered in the SI but does not really have much effect beyond 'resetting' the simulation (effectively back to $t=1$); if strategies had been competing then this would have more interesting effects, such as reducing the fitness of social learners as they copy out-dated information from each other. In sum, I think the setup here is less realistic than allowing strategies to directly compete, and has less relevance to both real human activities and the evolution of these strategies. This should be recognised. Statements such as "in order to increase individual and group-level performance" (p.2) or "solutions that improve individual and group-level performance" (p.3) are also misleading, as here you are only looking at group-level performance (i.e. the average performances plotted in the graphs) and not individual-level performance, because there is no individual variation in strategy.

4. I did not understand why random-copying did not perform well. It is argued that conformity did well (in efficient networks and complex landscapes) because it was inefficient, inhibiting exploitation of sub-optimal solutions and allowing longer exploration of the fitness landscape. But surely random copying would do exactly the same, if not more so. It is said that "since [random copying] is not biased towards any criteria related to success (such as the best option or the most frequent option) it is not able to drive the population to good solutions" (p.6). But surely this is incorrect, given that agents only switch to the random agent's solution if it is better than their existing solution (i.e. a 'copy-if-better' strategy). I would like to see more explanation of why this did not out-perform conformity, as it seems to contradict the explanation given for the success of conformity.

5. Please include source code for the simulations as SI, so others can replicate the models/results.

Minor points:

- references 1-10 do not seem to be cited in the text

Reviewer #3 (Remarks to the Author)

This is a useful paper, illustrating the effects of interaction between social learning strategies and network structure on performance. I have only minor comments for consideration:

1. Use of the term 'collective' in collective problem solving.

Individuals are networked allowing opportunity for social learning, so the term 'collective' seems redundant. You're simply dealing with individuals that are not isolated from one another.

2. Use of terms 'inefficient' and 'efficient' to describe networks

The authors define these in relation to the speed at which information spreads in the population. Surely, this is a function not only of network structure, but also the social learning strategies employed. Perhaps these are technical terms used by network theorists (?) to refer to the default case where individuals engage in unbiased social learning? It would be interesting to see an analysis of the rate of spread across the full range of specified networks (rather than just the steady state performance).

3. Specific parameter values are used to illustrate qualitative effects and in some cases it isn't clear how generalisable these are. $s=9$ is described as relatively large, but given $d=19$, s could be considerably larger. Have the authors looked at larger values of s , and are the results qualitatively consistent with the case where $s=9$? N is set at 100. Do the qualitative effects persist across a wide range of N ? Clearly it is not reasonable to assume that the same network structures will occur across a wide range of N .

4. When learning socially, individuals have the capacity to identify and copy the 'best' solution. Yet, when they're learning on their own, they appear to be stupid and modify their behaviour at random. If this is the case, the authors should justify this discrepancy.

5. On page 10, the discussion of Mason and Watts (2012) appears not to distinguish conformist bias from unbiased social learning: increase likelihood of copying a task that is exhibited by 2 or more demonstrators is expected under unbiased social learning. If there is frequency dependent learning, it isn't clear that it would be conformity (using Boyd and Richerson's sigmoidal definition): perhaps there's more information in M&W to help clarify?

6. For the discussion, it may be worth noting that individuals can be plastic in their use of social learning strategies and that perhaps network structure could influence the social learning strategy that is employed. Equally, non-random attention to members of the population may influence network structure. The authors fix network structure, which is useful to identify the observed effects and might relate to external fixation at a higher, institutional level. By contrast, dynamic interaction between behaviour and network structure could be consistent with structuration and practice theory (e.g. Giddens, Bourdieu).

Response to reviewers

Please find out point-by-point responses starting with ******* in italic.

Reviewers' comments:

Reviewer #1 (Remarks to the Author):

This is an excellent paper that addresses the utility of different social learning rules. It is very clearly written, and the results are convincing. I recommend that the paper be accepted as is.

I have one suggestion that the authors may want to consider. On page 6 the authors describe the random social learning rule as not being biased toward success. Readers who come from the evolution of social learning literature will find this confusing. Random social learners choose a person to learn from (a "model") at random, and then if the behavior that they learn from that person results in a higher payoff, they adopt the socially learned behavior. This is what social learning theorists like Joe Henrich and Richard McElreath call "payoff biased" learning, and it leads to replicator dynamic spread of the higher payoff behavior. It is also the model used by economists interested in social learning (e.g. Karl Schlag) and economists and sociologists who study the diffusion of innovations. The search process is not success biased, but taken over the whole cycle (choose model, learn behavior, decide whether to adopt) the rule is definitely biased toward success because only superior behaviors are adopted. It is very interesting that it fares so poorly compared to a conformist search rule, but the authors might consider adding a few sentences to clarify those who come to this paper with a different conceptual structure.

******This is a very good observation. The reason why random copying is performing so poorly is because we implemented it as a purely exploitation-based strategy (i.e. it does not engage in any individual learning). This strategy serves as a benchmark to see how far one can get with pure exploitation (while the pure individual learning strategy shows how far one can get with pure exploration). For the task we studied it is essential to balance the two types of behaviors. Pure replicator dynamic spread of the high payoff behavior (without any individual learning) quickly drives the group to a local peak. We now make this clear in the description of the strategies that this strategy engages only in exploitation (see page 3).*

A version of the random copying strategy that also engages in individual learning would be equivalent to the best member strategy with a sample size of $s=1$. We now include results for this strategy in the Supplementary Information (see Supplementary Table 2) where we explore the effect of different sample sizes (varying s). It indeed does perform well, slightly better than the best member ($s=3$) strategy, but it does not change any of the overall conclusions so we keep

the $s=3$ strategy in the main text in order to be comparable and to remain consistent with Mason and Watts (2012).

Reviewer #2 (Remarks to the Author):

This paper presents a simulation of different social learning strategies (e.g. copy the best other agent, or conform to the most common solution amongst n agents) in different fitness landscapes (a simple landscape with a single optimum, or a complex NK multi-peaked landscape) and within different social networks (highly connected vs sparsely connected). It is found that these 3 features interact. In simple landscapes all strategies perform equally well, but in complex landscapes conformity does better because it prevents early convergence on sub-optimal peaks (i.e. increases exploration). Also, conformity does better in efficient networks, but copy-best-agent does better in inefficient networks. The latter effect occurs for the same reason: copy-best's efficiency is reduced by the inefficient network, thus preventing early convergence on sub-optimal peaks. It is claimed that these interactions explain previous apparently-contradictory experimental findings.

Overall I like the study, particularly its claim to resolve conflicting experimental findings. This would be a valuable contribution to the literature. I think it does this only partially however. I have the following concerns/recommendations:

1. It is a problem for me that only in the SI is it mentioned that Mason & Watts actually used a different fitness landscape to the one analysed in the main text. This is a problem because the central claim of the paper is to resolve specifically Mason & Watts' findings with Lazer & Friedman's opposite findings (e.g. from p.9, figure caption: "The left panel replicates the findings of Lazer & Friedman (2007) while the right panel replicates the findings of Mason & Watts (2012)"). Figures S1-S4 show that when Mason & Watts' landscape is used, the effect is much less striking: conformity is still better in efficient networks, but copy-best shows little difference between efficient and inefficient. I would encourage the authors to be explicit about this in the main text, and ideally present these more representative results there rather than in the SI, if the authors wish to claim a link to those earlier studies.

******We agree with the observation that it is essential to state that these two studies focused on different landscapes. We thought carefully about the best way to present these results in a concise manner. We modified the sentences suggested above replacing "replicates" with "are in line with" and highlighted the fact that Mason and Watts studied a different landscape (see Page 6 bottom and Figure 3 legend). We also updated the networks we studied (see further explanation below the next point (2.)) and modified the individual learning rule in the Mason and Watts landscape to have a single learning rule instead of 4 different variants as originally (see Supplementary Note 3). Note that Mason and*

Watts in their Supplementary Information studied a number of different learning strategies in their attempt to reconstruct the inefficient network results, however, they did not succeed. Therefore, we propose a new learning rule (by combining their different variants of myopic search) that does replicate the inefficient network result. We also highlight that Mason and Watts only iterated their study for 15 time steps, which is insufficient time to judge the long-run effect of different networks. These changes made the results stronger, and we believe they are now qualitatively quite similar to the NK results, so we feel justified in reporting it only in the Supplementary Information. We are open to including it in the main text, however, due to space limitations and the size of the figures we could not find a good way to include it there.

2. Also relating to this comparison, the best-member data presented in Figures 3 and 4 seems to be highly skewed by the lattice outlier. Why is this, and can the authors assure the reader that the effect in the top-left of figure 3 is not simply driven by this outlier? Also, the regression lines in Figure 4 would be far more convincing if there were an equal number of data points for each diameter. Currently there are, for example, three data points for diameter=3, and none for diameter=7 or 8. Please consider increasing the number of diameter values.

******We thought carefully about this issue. We realized that since our networks are much larger than the ones studied by Mason and Watts (2012) the rewiring process needs to be run for considerably more repetitions to really converge. This is computationally quite expensive but we regenerated the network with 1.000.000 rewirings and 1000 repetitions (described on page 11). We now report a figure of the networks (Figure 3 in the Supplementary Information) which shows that we obtained the same, only larger network structures as in Mason and Watts (2012). The results with the new networks no longer show the lattice network as an outlier. Instead, the most extreme networks are the ones that have many isolated clusters (max-mean-clustering, max-var-constraint, see Figure 3 in the Supplementary Information).*

******Regarding the point about diameter: we agree that it would be ideal to have the same number of networks for each diameter value. However, diameter is an emergent property of the network generation process, and we know of no clean way of manipulating it without affecting other properties. We follow the analyses of Mason and Watts (2012) who report a similar analysis also with an uneven number of diameter values (see page 14 of their Supplementary Information) (We do mention this as an issue on page 7).*

3. I would like to see more connection to the evolution of social learning literature, which is highly relevant here. For example, citation and discussion of models of copy-best and conformity (see Laland, 2004, Learning & Behaviour). This has important implications for many of the claims. For example, here the authors are simulating populations in which every member has the same strategy

(e.g. all use conformity), measuring performance, then comparing that performance across populations of pure strategies. This is a purely group-level comparison. It assumes that each strategy's performance is independent of the presence of other strategies. The authors cite Rogers (1988) but do not seem to appreciate his result, which is that strategy performance is often not independent of others; they are often frequency dependent. For example, social learners do well amongst large numbers of individual learners because they can scrounge their information at low cost, while social learners do poorly when social learners are common because they cannot track environmental change. The present study cannot make such comparisons because at no point were strategies interacting within the same population. Environmental change is considered in the SI but does not really have much effect beyond 'resetting' the simulation (effectively back to $t=1$); if strategies had been competing then this would have more interesting effects, such as reducing the fitness of social learners as they copy out-dated information from each other. In sum, I think the setup here is less realistic than allowing strategies to directly compete, and has less relevance to both real human activities and the evolution of these strategies. This should be recognised. Statements such as "in order to increase individual and group-level performance" (p.2) or "solutions that improve individual and group-level performance" (p.3) are also misleading, as here you are only looking at group-level performance (i.e. the average performances plotted in the graphs) and not individual-level performance, because there is no individual variation in strategy.

***** We have removed all references to "individual and group-level performance" and replaced it with "group-level performance". We agree with all other points raised and devote a separate section in the discussion to them (see page 10).*

4. I did not understand why random-copying did not perform well. It is argued that conformity did well (in efficient networks and complex landscapes) because it was inefficient, inhibiting exploitation of sub-optimal solutions and allowing longer exploration of the fitness landscape. But surely random copying would do exactly the same, if not more so. It is said that "since [random copying] is not biased towards any criteria related to success (such as the best option or the most frequent option) it is not able to drive the population to good solutions" (p.6). But surely this is incorrect, given that agents only switch to the random agent's solution if it is better than their existing solution (i.e. a 'copy-if-better' strategy). I would like to see more explanation of why this did not out-perform conformity, as it seems to contradict the explanation given for the success of conformity.

****** Please see our response to Reviewer #1.*

5. Please include source code for the simulations as SI, so others can replicate the models/results.

****** All our code is now available at*

https://www.github.com/dnlbrkc/social_learning

Minor points:

- references 1-10 do not seem to be cited in the text

****** Thank you for noticing this. This is now addressed.*

Reviewer #3 (Remarks to the Author):

This is a useful paper, illustrating the effects of interaction between social learning strategies and network structure on performance. I have only minor comments for consideration:

1. Use of the term 'collective' in collective problem solving. Individuals are networked allowing opportunity for social learning, so the term 'collective' seems redundant. You're simply dealing with individuals that are not isolated from one another.

***** We now removed all references to the term 'collective'.*

2. Use of terms 'inefficient' and 'efficient' to describe networks
The authors define these in relation to the speed at which information spreads in the population. Surely, this is a function not only of network structure, but also the social learning strategies employed. Perhaps these are technical terms used by network theorists (?) to refer to the default case where individuals engage in unbiased social learning? It would be interesting to see an analysis of the rate of spread across the full range of specified networks (rather than just the steady state performance).

****** We agree that the terms 'inefficient' and 'efficient' are not very useful as they are quite vague. However, we chose the use these terms because they were used in the studies that we aimed to reconcile (Lazer and Friedman, 2007; Mason and Watts, 2012). The diameter of a network (i.e., its path length) is a measure of how fast information can diffuse in the case of unbiased information transmission so it is a clear indicator of speed of diffusion.*

3. Specific parameter values are used to illustrate qualitative effects and in some cases it isn't clear high generalisable these are. $s=9$ is described as relatively large, but given $d=19$, s could be considerably larger. Have the authors looked at larger values of s , and are the results qualitatively consistent with the case where $s=9$? N is set at 100. Do the qualitative effects persist across a wide range of N ? Clearly it is not reasonable to assume that the same network structures will occur across a wide range of N .

****We explore the effect of different sample sizes in the Supplementary Information, ranging from the smallest to the largest possible sample size (Supplementary Table 3) and find that they do not change any of the results. We stick to a sample size of $s=3$ in the main text as this is also the sample size that participants in Mason and Watts (2012) had access to. Unfortunately, we are not able to explore several different values of N given the computational costs of generating larger networks, however, network-based studies of social learning, cooperation and economic games typically all focus on a single value of N , assuming that results scale up (see recent studies by Duncan Watts, Nicholas Christakis, Maxime Derex and Robert Boyd). In the present case we can safely assume that results scale, since previous studies have investigated different group sizes (Derex & Boyd: $N=6$; Mason & Watts: $N=16$; Lazer & Friedman: $N=100$; Wisdom, Jones & Goldstone: $N=9$). Also from Figure S8 in the Supplementary Information we can see that our network structures look quite similar to the smaller networks studied in Mason and Watts (2012).

4. When learning socially, individuals have the capacity to identify and copy the 'best' solution. Yet, when they're learning on their own, they appear to be stupid and modify their behaviour at random. If this is the case, the authors should justify this discrepancy.

**** We now cite several sources that use this strategy (see page 3). It is a common myopic search strategy in the literature on rugged landscapes (e.g., Levinthal, 1999; Lazer and Friedman, 2007), often referred to as hill-climbing. It is one of the most effective search strategies also studied by the author of the NK landscape (Kaufmann, 1993).

5. On page 10, the discussion of Mason and Watts (2012) appears not to distinguish conformist bias from unbiased social learning: increase likelihood of copying a task that is exhibited by 2 or more demonstrators is expected under unbiased social learning. If there is frequency dependent learning, it isn't clear that it would be conformity (using Boyd and Richerson's sigmoidal definition): perhaps there's more information in M&W to help clarify?

****We really like this point and thought about it carefully. We agree that both unbiased and conformist social learning would show an increased chance of copying the most frequent solution, however, Figure 4 Panel A in Mason and Watts (page 4.) shows a disproportionate tendency to copy the most frequent solution. According to Efferson et al (2007) this disproportionate tendency is what distinguishes conformist from unbiased social learning. We now cite this paper in the discussion and use the term 'disproportionately' (page 9)

6. For the discussion, it may be worth noting that individuals can be plastic in their use of social learning strategies and that perhaps network structure could influence the social learning strategy that is employed. Equally, non-random attention to members of the population may influence network structure. The

authors fix network structure, which is useful to identify the observed effects and might relate to external fixation at a higher, institutional level. By contrast, dynamic interaction between behaviour and network structure could be consistent with structuration and practice theory (e.g. Giddens, Bordieu).

***** *We fully agree and now devote a new section to this in the discussion (page 10).*